# The Cost of Imagined Actions in a Reward-Valuation Task

**DOI:** 10.3390/brainsci12050582

**Published:** 2022-04-29

**Authors:** Manuela Sellitto, Damiano Terenzi, Francesca Starita, Giuseppe di Pellegrino, Simone Battaglia

**Affiliations:** 1Centre for Studies and Research in Cognitive Neuroscience, Department of Psychology, University of Bologna, 40126 Bologna, Italy; manuela.sellitto@psypec.it (M.S.); francesca.starita2@unibo.it (F.S.); 2School of Psychology, Bangor University, Bangor LL57 2AS, UK; 3Department of Decision Neuroscience and Nutrition, German Institute of Human Nutrition (DIfE), 14558 Potsdam-Rehbrücke, Germany; damiano.terenzi@dife.de; 4Charité—Universitätsmedizin Berlin, Corporate Member of Freie Universität Berlin, Humboldt-Universität zu Berlin, Berlin Institute of Health, Neuroscience Research Center, 10117 Berlin, Germany

**Keywords:** delay discounting, effort discounting, Fitts’ law, motor imagery, mental simulation, reward value, visual imagery

## Abstract

Growing evidence suggests that humans and other animals assign value to a stimulus based not only on its inherent rewarding properties, but also on the costs of the action required to obtain it, such as the cost of time. Here, we examined whether such cost also occurs for mentally simulated actions. Healthy volunteers indicated their subjective value for snack foods while the time to imagine performing the action to obtain the different stimuli was manipulated. In each trial, the picture of one food item and a home position connected through a path were displayed on a computer screen. The path could be either large or thin. Participants first rated the stimulus, and then imagined moving the mouse cursor along the path from the starting position to the food location. They reported the onset and offset of the imagined movements with a button press. Two main results emerged. First, imagery times were significantly longer for the thin than the large path. Second, participants liked significantly less the snack foods associated with the thin path (i.e., with longer imagery time), possibly because the passage of time strictly associated with action imagery discounts the value of the reward. Importantly, such effects were absent in a control group of participants who performed an identical valuation task, except that no action imagery was required. Our findings hint at the idea that imagined actions, like real actions, carry a cost that affects deeply how people assign value to the stimuli in their environment.

## 1. Introduction

What is the relation between the value that the brain assigns to a rewarding stimulus and the cost of the action required to obtain it? The value an agent gives to a reward is well known to be determined by its immediate sensory characteristics (e.g., the amount of a food product or its taste), as well as by individual preferences and experiences. However, among the majority of animals, the decision to forage often implies the computation of benefits against predictable costs, including the effort required to obtain the reward [1,2,3,4]. Recent behavioural and neuroeconomic studies have suggested that the value assigned to a reward stands in inverse relation to the amount of effort required to obtain it [5,6,7,8]. In other words, people give more value to those rewards that can be achieved with less effort [9,10,11]. Such effort can be physical but also cognitive (e.g., by manipulating visuospatial task demands) [12]. Therefore, it is plausible to expect that value computation can be influenced by the degree of cognitive effort as well [13,14].

Similar to effort, another cost that is often incurred during decision-making is the delay that one has to endure before receiving a reward. This phenomenon is known as temporal discounting (TD) and reflects the decrease in the subjective value of a reward [15,16,17,18,19]. Recently, TD has been deemed to affect the motor commands responsible for the movements of the human body necessary to reach a stimulus [20,21,22]. When a stimulus is associated with a future reward, lower dopaminergic neurons firing is observed, as well as slower saccades for the visual reaching of the stimulus are recorded [23,24]. In addition, when a stimulus is associated with an immediate reward, higher dopaminergic neuron firing usually precedes faster movements [25,26]. Therefore, the kinematics of movements seem to reflect the processes with which the brain temporally discounts rewards, suggesting a link between motor and valuation processes [20,27,28]. Movements and time share a well-known relation as well: according to Fitts’ law [29], the time needed by an individual to move between a starting point and a target point within a given space increases linearly with the difficulty to execute the movement. In addition, the time for mentally simulating movements (motor imagery, MI) [30,31,32,33,34,35] is highly correlated with the time to actually make such movements. Thus, mental movements closely mimic real movements in their temporal organization, involve the same planning programs [36] and rely on similar neural processes [37,38,39]. Furthermore, Fitts’ law accounts equally well for imagined and executed movements [40,41,42,43].

A substantial body of neurophysiological and neuroimaging studies suggests that there are brain areas and neural populations responsible for both the decisional process and the selection of an action [44,45,46,47]. These areas represent a common substrate for the different kinds of decision-making processes. In particular, the orbitofrontal cortex (OFC) has a critical role in representing the subjective value of a reward, thus employing such value to condition the choice among different options [5,48,49,50]. In addition to the OFC, value is represented in other regions, including those related to motor preparation and execution [51,52,53,54,55,56]. In line with these studies, decisions based on value would affect the competition between representations of actions and the movement itself before the decision-making processes are completed [57,58]. Moreover, action representation has also been observed to influence the value of an item during the decision-making processes. Such influence could be interpreted as a sort of value signal from the motor and pre-motor regions [44,55,59,60,61]. Based on this evidence, it can be argued that the mechanisms underlying the computation of reward value are extremely sensitive to the cost of an action needed to obtain such reward. Importantly, the preference for an action is inversely proportional to its effort, even in terms of time. This can, in turn, influence the preference of a reward such that when the action is more difficult and requires more time to be executed, the reward may be perceived as less valuable.

In the present study, we investigated, for the first time, whether imagined movements, like real ones [23,62], would carry a cost that affects how people assign value to food items. These processes are currently unexplored, and they can be useful in better understanding diseases such as apathy, which is characterized by an alteration of abilities to anticipate the effort or difficulty. To this end, via a novel experimental procedure, participants performed an experimental task in which the pictures of food and drink item were associated with two different path widths (large or thin) that were really and imaginarily navigated via a mouse cursor. After navigation, participants rated how much they liked and wanted those items on a Likert scale. We expected participants to assign a higher value score (i.e., wanting and liking) to food items that required a shorter time (lower cost) to be reached (large path), relative to those items requiring a longer time (higher cost, thin path). Thus, thin path was expected to take more time, incurring a higher cost, while large path was expected to require less time and hence lower cost.

## 2. Materials and Methods

### 2.1. Participants

A power analysis based on previously published studies [63,64,65,66] indicated that a sample size of ~20 participants was necessary to achieve a statistical power of >95% (2-tailed = 0.05). Thus, a total of 40 young adult right-handed volunteers (22 female) with a mean age of 23.6 years (sd = 2.76) and mean education of 16.87 years (sd = 2.16) were recruited from the student population of the University of Bologna for a single-session experiment (see Table 1 for further demographic information). Participants were randomly assigned either to the experimental group or to the control group. Body Mass Index (BMI) was calculated for each participant and then overall for each group [67]. Moreover, hunger (on a 5-point Likert scale ranging from 0 “not at all hungry” to 5 “extremely hungry”) and fasting level (hours without eating) were collected. Subjects remained naïve as to the purpose of the study until debriefing at the end of the experimental session. The two groups did not show any significant difference in terms of BMI index, hunger and fasting level (all ps > 0.2). Participants had no history of psychiatric or neurological diseases, had normal or corrected-to-normal vision and all gave written informed consent before the beginning of the experiment. The experimental protocol was conducted in accordance with the Declaration of Helsinki (2008) and approved by the Institutional Review Board (or Ethics Committee) of the University of Bologna (protocol number 14, 8 February 2013).

### 2.2. Apparatus and Stimuli

The experiment was implemented in E-prime 2.0 (Psychology Software Tool, Sharpsburg, PA, USA) and run on a Windows-based PC (Lenovo ThinkCentre Desktop Computer, Beijing, China). Participants sat in front of a 15-inch colour LCD monitor (1024 × 768 pixels) with an unconstrained viewing distance of approximately 65 cm.

Two rectangles were displayed centred on the screen in two different positions. One rectangle (55 × 75 mm) was positioned in the upper part, and the other (15 × 75 mm) in the lower part of the screen. They could be connected either through a large (155 × 65 mm) or a thin (155 × 20 mm) path (Figure 1). These paths induced participants to perform a potential movement over the path. In particular, the thin path forced participants to perform more careful movements compared to the large one [68]. Pictures of food and drink items (50 × 50 mm) appeared one at the time inside the top rectangle, while the bottom one was used as the starting point. The size of the paths was randomized among trials (see Figure 1). Pictures of Food and drink items were coloured pictures of 28 different snack food and drink items consisting of candy bars (e.g., Bounty or KitKat), potato crisps (e.g., Fonzies), crackers (e.g., Ritz), fruit juices (e.g., Pago) and carbonated drinks (e.g., Sprite). Such items were available at local convenience stores, each with a mean price of EUR 1.36 (sd = 0.54). To reduce the influence of personal food preference, the presentation of the stimuli was counterbalanced across participants. All stimuli were balanced for luminance, complexity and colour saturation using photo editing software (Adobe Photoshop CS6).

### 2.3. Tasks

#### 2.3.1. Experimental Valuation Task

In this task, and for each trial, participants first had to answer to two questions. Second, they had to imagine themselves dragging the mouse along the path shown on the screen (thin or large) from the starting point rectangle to the upper rectangle. Specifically, they were required to imagine keeping the mouse cursor inside the path until the food item was reached.

Trials started with a black fixation cross, centrally displayed on a white background (1000 ms), followed by the presentation of a food item, together with one of the two paths. Participants could observe the screen for as long as they needed. When they felt ready, they were instructed to press the spacebar so that one question and its rating scale appeared at one bottom corner of the screen. The order and screen position of questions and their respective scales were counterbalanced across participants. The two questions (in Italian) were “How pleasant would it be to experience eating this item now?” (liking ratings), and “How much do you want to eat it now?” (wanting ratings) [69]. Participants answered through a seven-point Likert scale (ranging from 1 “not at all” to 7 “extremely”) [70] by pressing the corresponding number on the keyboard. After the last rating was submitted, the question and the scale faded out. Then, mouse cursor appeared, centred within the bottom starting rectangle. At this point, participants had to imagine, in a first-person perspective, dragging the mouse along the path from the starting point to the food item picture in the upper rectangle at the end of the path. They had to press the left mouse button with the index finger of the right hand to signal the start of their imagined movement, and the button had to be pressed again to flag the imagined reaching of the food item. Trials were separated by a 1000 ms inter-trial interval (ITI) with a blank screen (see Figure 2). The experimental valuation task consisted of 56 randomized trials, 28 for each path width. Before starting the task, participants had to perform three trials for training. They had to press the left mouse button with the index finger of the right hand, drag the mouse inside the path and reach the food item. Finally, the mouse button had to be pressed again to complete the trial. It is important to note that during the training, participants had to physically move the mouse along the path to ensure that participants understood the kinetic characteristics of the actions [68]. Indeed, the training aimed at familiarising participants with the reaching movement that they had to imagine completing across the paths, where the large path was assumed to be easier than the thin one.

#### 2.3.2. Control Valuation Task

The control valuation task was designed to ensure that only the imagery time required to perform the reaching movement affected the value assigned to the item, as performed during the experimental task. Before participants began the control valuation task, a short training (e.g., 3 trials) was performed, during which they were familiarised with the food and drink items, as well as with the two different paths, to ensure that they could understand the characteristics of the task (e.g., the different paths). After the initial set of training trials, participants started the task. The sequence of trial events was equivalent to the experimental task. There was a 1000 ms fixation period, followed by the presentation of a food item and one of the two paths. They could observe the path on the screen for as long as they wanted and, as in the experimental task, they were provided with liking and wanting rating scales. Subsequently, participants had to press the left mouse button twice, as done in the experimental task, although here it had no relation with path navigation. Finally, a 1000 ms ITI blank screen appeared. Notice that, differently from the experimental task, participants were not requested to make or imagine any action.

### 2.4. Experimental Procedures

The study was performed at the Centre for Studies and Research in Cognitive Neuroscience (CsrCN), Cesena, Italy. Participants sat comfortably in a silent room, and their demographic data, including height and weight with self-report questionnaires, were collected in order to calculate their BMI. They were also asked to rate their hunger at the moment of the experiment and estimate their fasting level (hours). In addition, participants filled out two questionnaires assessing imagery abilities: The Vividness of Visual Imagery Questionnaire (VVIQ; [71]) and the Vividness of Movement Imagery Questionnaire (VMIQ [72]). The main purpose of these questionnaires was, on the one hand, to assess participants’ ability to imagine movements. On the other hand, they raised the vividness of the motor imagination, which was necessary for the experimental task. The VVIQ consists of 16 items, divided into four groups of four items each, in which participants are invited to vividly imagine specific scenes and situations, e.g., a sunrise evolving into a rainstorm. The VMIQ consists of 24 items asking participants to describe a simple movement, such as walking, or a complex movement, such as rope jumping. The VVIQ and VMIQ consist of items triggering the vivid imagination of movement, rated on a five-point scale. The order of questionnaire submission was counterbalanced across participants. After filling out the questionnaires, participants performed the task. At the end of the session, they were debriefed on the purpose of the study.

### 2.5. Statistical Analyses

The aim of this study was to investigate how the effort, associated with imagining moving along each path, influenced the value assigned to the food and drink items. The significance of the experimental factors was tested using 2 × 2 ANOVAs—one for the liking rating, one for the wanting rating and one for the reaction times as dependent variables—using group (experimental, control) as the between-participants independent variable and the path width (large, thin) as the within-participants independent variable. The alpha-level of all analyses was set at *p* < 0.05 using a univariate approach. Furthermore, scores related to the questionnaires were analysed. All statistical analyses were performed with SPSS Statistics (IBM Corp, released March 2013, IBM SPSS Statistics for Windows, version 22.0, Armonk, NY, USA) and STATISTICA (Dell Software, released September 2015, StatSoft STATISTICA for Windows, version 13.0, Round Rock, TX, USA).

## 3. Results

### 3.1. Liking

The ANOVA performed on the liking rating revealed a significant main effect of the path width (F(1, 38) = 22.461, *p* < 0.001, part.η^2^ = 0.4), a significant group X path width interaction (F(1, 38) = 5.958, *p* = 0.019, part.η^2^ = 0.1; Figure 3) and no significant group effect (*p* = 0.778). Newman–Keuls post-hoc analysis revealed a significant difference (*p* < 0.001) between the large path (mean = 4.74; sd = 0.89) and the thin path (mean = 4.15; sd = 0.88) in the experimental group, but not in the control group (*p* = 0.112; large path: mean = 4.62, sd = 0.95; thin path: mean = 4.43, sd = 1.02). These results indicate that the path widths significantly influenced the liking ratings only in the experimental group, with greater liking when items appeared at the end of the large path than the thin path.

### 3.2. Wanting

The ANOVA performed on the wanting rating revealed a significant main effect of the path width (F(1, 38) = 8.219 *p* = 0.006 part.η^2^ = 0.17), but no significant group X path width interaction (F(1, 38) = 0.259 *p* = 0.613) and no group effect (*p* = 0.281; Figure 4). These results indicate that the path widths influenced the wanting rating such that for the large path (mean = 3.44; sd = 1.02), participants showed higher wanting score compared to the thin path (mean = 3.22; sd = 1.04), regardless of the group.

### 3.3. Response Time

The ANOVA performed on the participants’ response times revealed a significant main effect of the path (F(1, 38) = 37.459, *p* < 0.001, part.η^2^ = 0.49), a significant main effect of the group (F(1, 38) = 34.360, *p* < 0.001, part.η^2^ = 0.47) and a significant group X path width interaction (F(1, 38) = 34.283, *p* < 0.001, part.η^2^ = 0.48; Figure 5). Newman–Keuls post-hoc analysis revealed a significant difference (*p* < 0.001) between the large path (mean = 3152.91 ms; sd = 2437.17 ms) and thin path (mean = 3994.39 ms; sd = 2665.35 ms) on the response times in the experimental group, but not in the control group (*p* = 0.852; large path trials: mean = 236.25 ms, sd = 140.93 ms; thin path trials: mean = 254.88 ms, sd = 166.54 ms). These findings indicate that the path width had a significant influence on response times only in the experimental group, with greater response times when imagining navigating the thin path than the large path.

### 3.4. Questionnaires and Correlations

The analysis of the questionnaires revealed that the experimental group’s VVIQ mean score was 58.4 (sd = 7.93), and control group’s VVIQ mean score was 63.9 (sd = 8.41) out of a maximum of 80. The experimental group’s VMIQ mean score was 84.7 (sd = 15.75), and control group’s VMIQ mean score was 89.3 (sd = 14.31) out of a maximum of 120 (Table 2). Two-tailed Pearson correlations were performed between the participants’ scores on the questionnaires and response times. The VVIQ scores negatively correlated with the response times (*r* = −0.363 (*p* = 0.021)), while no correlation resulted for the VMIQ scores (*r* = −0.17 (*p* = 0.283)). The significant negative correlation suggests that lower response times were associated with higher scores of vividness of visual imagery but not of motor imagery.

## 4. Discussion

Although different studies have investigated multiple factors involved in value computation [47,52,73,74,75,76], including risk [77,78], spatial distance [79,80,81], effort [9,10,11,12,82] and temporal discounting [15,16,17,18], little is known about how the cost of an action impacts on valuation processes. This study describes a new experimental protocol for studying and quantifying how such cost can affect valuation processes and, thus, behaviour. In particular, this research explores how the subjective value assigned to an item can be affected by manipulating the difficulty of an imaged action to reach it.

A previous study showed that the cost of real actions could affect how people assign value to the stimuli in their environment [23]; however, to the best of our knowledge, no study to date has tested the cost of imagined actions. To this end, first, participants familiarized with physically moving the mouse cursor inside each of two different paths. Second, they were asked to imagine dragging the mouse inside these paths in order to reach a food item. According to Fitts’ law [29], given the different path widths, participants’ imagined movements implied a variable difficulty in reaching the target; such difficulty was revealed by the discrepant timings needed to navigate each path. In addition, participants answered two questions via a seven-point Likert scale, which was designed to evaluate the liking and wanting of each food item. Participants took shorter time to imagine navigating the large path than the thin path. In addition, they liked more foods and drinks at the end of the large path than the thin path, but only if they were performing the motor imagery task. These results suggest that the cost (in terms of time) of the imagined action influences value ratings, with less value attributed to items that involve higher action cost. The rating differences were obtained by increasing or decreasing the path widths and, thus, manipulating the cost of the imagined action required to reach the item. The higher liking to the lower-difficulty and short-time items at the end of the large path reveals that an action’s cost is computed even before its execution [83,84,85]. Furthermore, these data show how the value of an item is influenced by the time required to reach it, even if the reaching is only imagined. In other words, presenting a food item with a large or thin path induced participants to anticipate the cost of the action, consequently affecting their evaluation. This is known as effort discounting [6].

The results reveal that an increased difficulty in mentally performing an action implied lower scores on the liking ratings for each item; the same expected result was not found across the wanting ratings. We explained that this discrepancy was caused by how the question may have been interpreted by participants. In line with this hypothesis, a recent systematic review of the human literature revealed a contradictory operationalization of the wanting and liking constructs [86]. Pool and colleagues [86] argued that expected pleasantness represents a major conceptual confound. In particular, expected pleasantness is an evaluation of how good or how bad a particular reward is going to be. This prediction involves the active reconstruction of past episodic memories of liking experiences with the current reward and the use of these episodic memories to anticipate or predict a future experience [87,88,89]. Whereas expected pleasantness drives cognitive desires, the interaction between the individual’s physiological state and the perception of the relevant reward-associated cue determines incentive salience or wanting [90]. Therefore, cognitive desires driven by memories of past liking experiences are not completely independent from liking, whereas wanting is underlain by a mechanism that is independent from the liking component [91,92,93]. Furthermore, the cognitive domain, including memory, is in constant complex interaction with other behavioural domains such as the negative valence domain, including depression and anxiety, which is under the control of endogenous neuropeptides, neurohormones and metabolites [94,95,96,97,98,99]. In addition, the functions of the domains are subject to the close influence of the adjacent biosystems [100,101,102,103,104].

In other words, it may be possible that our experimental manipulation had an effect on the expected pleasantness of eating or drinking the items, which is more related to self-reported measures of liking rather than wanting. In this regard, it has been shown in the human literature that self-reported rating scales seem to be the most reliable index of the hedonic experience [105,106]. Therefore, our results suggest that the cost of an imagined action directly influences the food item’s value computation, even though the action is not related to the hedonic features of such item. Specifically, the difficulty to reach an item is also included in the computation of value, in addition to the main sensory features that define that item. Crucially, in this study, the participants’ rating was affected because they were anticipating the specific difficulty of each action while performing the experimental trials. Our results support the hypothesis that imagining actions with variable difficulties automatically influences the item value.

Concerning response time, the results showed that the experimental group took more time to imagine making a movement along the thin path rather than the large path. These data revealed that simply imagining such movement influenced the participants’ response time, as well as their ratings. As previously described, we also found some correlations. The significant negative correlations between the VVIQ score and response time revealed that an increased participants’ vividness of imagination resulted in a reduced motor imagery time and, therefore, in reaching the food item faster.

These findings indicate that the value of the item is influenced by the cost of the action required to obtain it. In line with previous studies, items that can be obtained with minor difficulty have a higher value because they can be reached more easily [9,10,92,107], and this occurs even when performing the reaching of the object is not necessary [108]. According to these data, the value of the reward is inversely proportional to the amount of effort required to reach it. In general, this evidence confirms the main principle that governs effort discounting: a higher effort leads to a devaluation of the reward.

Our results could be interpreted according to an evolutionary perspective, which makes plausible the interaction between the computation processes underlying the value of an item and those codifying the difficulty to obtain it. Therefore, our evidence could be included along the theories of embodied cognition [109,110,111]. These theories emphasize the role of the body in cognition, arguing that the body’s condition influences cognitive states as they, at the same time, influence the body. According to these theories, participants’ knowledge of the world would be represented by neural structures which store sensory, motor and affective information. In agreement with this idea, the simple reaching of an object, or anticipating the difficulty to obtain it, activates such neural structures which are also involved in one’s own perceptual experience. In line with our results, the anticipation of an action’s difficulty could be considered as a simulation of the effort required for executing the action. The process governing the value computation could thus be influenced by the physical or cognitive effort required to reach it. Our data are further confirmed by, and in line with, several neurophysiological and neuroimaging data. These data show a possible representation of the subjective value not only in areas directly related to value computation and decision-making processes, such as ventromedial prefrontal cortex (vmPFC) [112,113,114,115,116,117,118,119], but also in other cortices related to motor preparation and action [51,52,120,121]. According to this evidence, the economic choice would be embodied in the motor processes of action selection [27,122].

## 5. Conclusions

In conclusion, our data show that the difficulty of an imagined action can influence valuation judgements. Moreover, our results support the hypothesis of the close relationship between motor and valuation processes. These links remain largely unexplored in current investigations, and they can be useful to better understand neurological or psychiatric conditions, such as schizophrenia, depression or frontal-apathy, which are characterized by an alteration of the ability to anticipate the effort necessary to obtain rewarding items in the environment [123,124,125,126,127,128]. Finally, future studies should also investigate the neural basis of this phenomenon; consequently, it would be very informative to submit a similar experimental paradigm to patients with focal brain lesions, for example, lesions affecting the medial orbitofrontal cortex (mOFC) or the anterior cingulate cortex.

## Figures and Tables

**Figure 1 brainsci-12-00582-f001:**
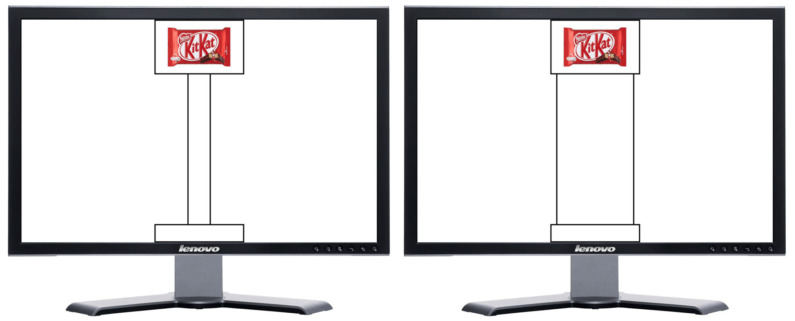
Examples of visual stimuli and paths. Shown brands for demonstrative purpose only.

**Figure 2 brainsci-12-00582-f002:**
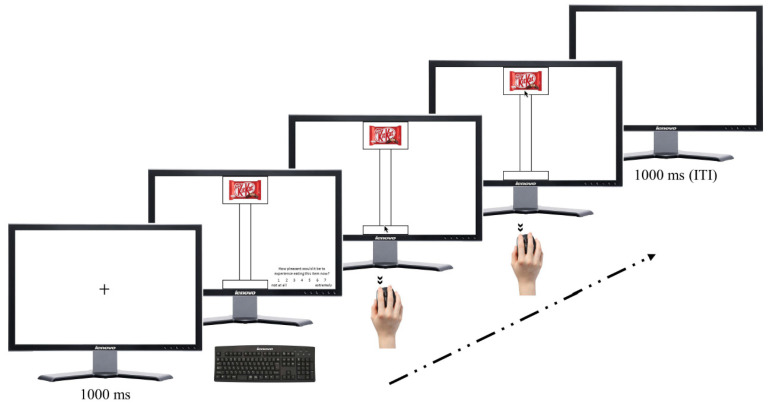
Schematic representation of the experimental trial sequences. Participants had to imagine, in a first-person perspective, dragging the mouse along the path, from the starting point to the food item picture in the upper rectangle at the end of the path. They had to press the left mouse button with the index finger of the right hand to signal the start of their imagined movement, and the button had to be pressed again to flag the imagined reaching of the food item. Trials were separated by a 1000 ms inter-trial interval (ITI) with a blank screen.

**Figure 3 brainsci-12-00582-f003:**
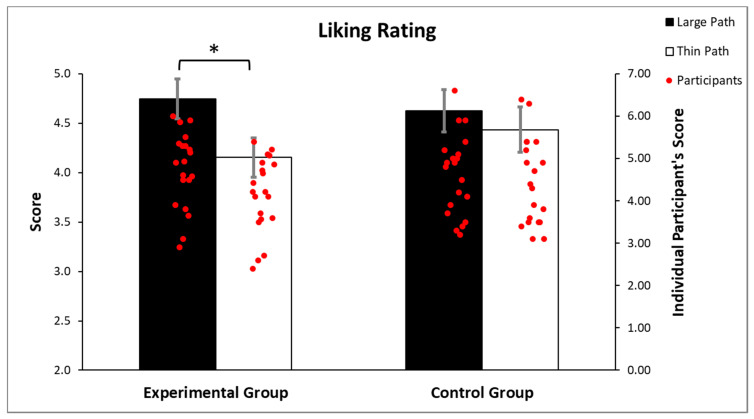
Analysis of the liking scores illustrates that the liking ratings were influenced by the path widths only in the experimental group (*p* < 0.001). Vertical lines indicate the standard error of mean (SEM), * *p* < 0.05.

**Figure 4 brainsci-12-00582-f004:**
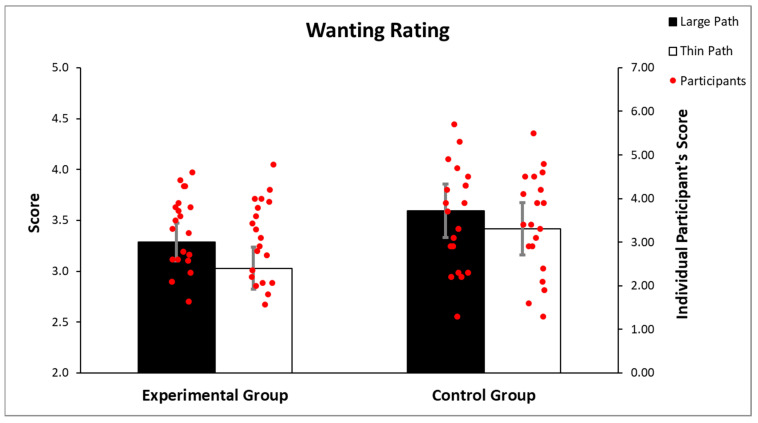
Analysis of the wanting scores illustrates that the wanting ratings were not differently influenced by the path widths in both groups. Vertical lines indicate the standard error of mean (SEM).

**Figure 5 brainsci-12-00582-f005:**
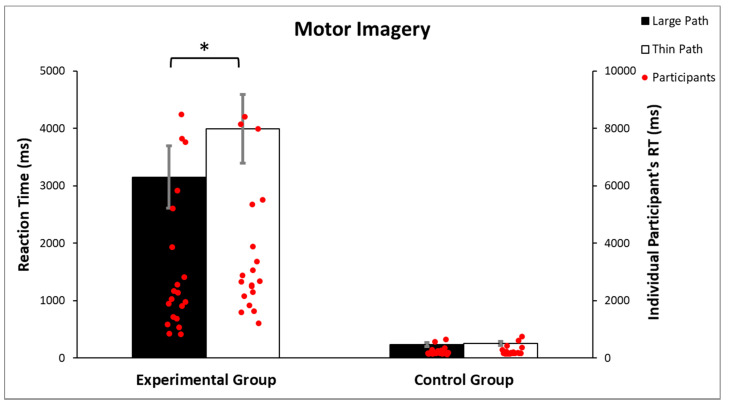
Analysis of the motor imagery time illustrates that the path width had an influence on the response time only in the experimental group. Vertical lines indicate the standard error of mean (SEM), * *p* < 0.001.

**Table 1 brainsci-12-00582-t001:** Participants’ demographic data.

Group	*n*	Age	Education	Hunger	Fasting	BMI
Experimental	20	23.35 (2.16)	16.8 (1.85)	2.25 (1.45)	2.08 (1.34)	21.49 (2.49)
Control	20	23.85 (3.36)	16.95 (2.46)	2.75 (1.16)	2.18 (1.32)	21.52 (3.16)

*n* is the number of participants. Age and Education are expressed in years. Hunger is expressed on a five-point scale. Fasting is expressed in hours. Body Mass Index (BMI) is expressed in kg/m^2^. Mean is represented out of the brackets, while the standard deviation is within them.

**Table 2 brainsci-12-00582-t002:** Participants’ questionnaire scoring.

	VVIQ		VMIQ	
**Group**	Experimental	Control	Experimental	Control
**Mean**	58.4	63.9	84.7	89.3
**SD**	7.93	8.41	15.75	14.31

The VVIQ score out of a maximum of 80, and the VMIQ score out of a maximum of 120.

## Data Availability

The datasets used and analysed during the current study are available from the corresponding authors on reasonable request.

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
