# Peer review of "The Cost of Imagined Actions in a Reward-Valuation Task"

_brainsci, 2022, doi:10.3390/brainsci12050582_

Round 1

Reviewer 1 Report

The article entitled: The cost of imagined actions in a reward-valuation task concerns an interesting cognitive subject. One should agree with the fact that, growing evidence suggests that humans and other animals assign value to a stimulus based on its inherent rewarding properties, but also on the costs of the action required to obtain it, such as the cost of time. Authors in this paper examined whether such cost also occurs for mentally simulated actions. Healthy volunteers indicated their subjective value for snack foods while the time to imagine performing the action to obtain the different stimuli was manipulated. In each trial, the picture of one food item and a home position connected through a path were displayed on a computer screen. 

My comments to the article are as follows:

- I propose to expand the sections of Motor Imagery also referring to research connecting the engine imagery and brain-computer technology. For example, I recommend referring to the article: Brain-computer technology based training system in the field of motor imagery, IET Science Measurement and Technology from 2020.

- Please, kindly provide arguments on the basis of which the research group was selected?

- In addition, please kindly provide information on what basis the ANOVA method was chosen for the data analysis? Has it been compared with other methods?

- I suggest, after the Discussion section, enter the Conclusions section. This section is currently missing. Some of the data from Discussion should appear in Conclusions. I also propose to refer to future plans for this research.

- The bibliography in the article is quite extensive, but often these are old items from 5 years ago. Some of them should be updated. For example, by referring to the article: refer to the article: Brain-computer technology based training system in the field of motor imagery, IET Science Measurement and Technology from 2020.

Author Response

Manuscript ID: brainsci-1706944 – Sellitto et al.,

We thank the Editor for returning our manuscript following peer review. We appreciate the positive and constructive feedback of the two Reviewers. In the present revision, we have dealt with all the points raised, and we feel the manuscript has significantly improved as a result.

REVIEWER #1 – MINOR REVISIONS
The article entitled: The cost of imagined actions in a reward-valuation task concerns an interesting cognitive subject. One should agree with the fact that, growing evidence suggests that humans and other animals assign value to a stimulus based on its inherent rewarding properties, but also on the costs of the action required to obtain it, such as the cost of time. Authors in this paper examined whether such cost also occurs for mentally simulated actions. Healthy volunteers indicated their subjective value for snack foods while the time to imagine performing the action to obtain the different stimuli was manipulated. In each trial, the picture of one food item and a home position connected through a path were displayed on a computer screen. 

Reviewer #1, comment 1:
I propose to expand the sections of Motor Imagery also referring to research connecting the engine imagery and brain-computer technology. For example, I recommend referring to the article: Brain-computer technology-based training system in the field of motor imagery, IET Science Measurement and Technology from 2020.

Authors’ reply to Reviewer #1, comment 1:
We would like to thank the Reviewer for this suggestion. Accordingly, the study by Szczepan Paszkiel and Paweł Dobrakowski and other relevant studies are now added in our manuscript when discussing Motor Imagery.

Reviewer #1, comment 2:
Please, kindly provide arguments on the basis of which the research group was selected?

Authors’ reply to Reviewer #1, comment 2:
Thanks for the question that allows us to clarify this point. The sample of participants in this study was recruited on a voluntary basis from the student population of the University of Bologna. In a blinded manner, the experimenters divided them into the experimental group and the control group without any assignment criteria. The number of the group was instead selected a priori based on a power analysis, please consider the answer to comment 1 of reviewer 2 for more details.

Reviewer #1, comment 3:
In addition, please kindly provide information on what basis the ANOVA method was chosen for the data analysis? Has it been compared with other methods?

Authors’ reply to Reviewer #1, comment 3:
We thank Reviewer 1 for their questions and concerns. Our chosen method of statistical data analysis depended on the fact that the data was normally distributed as well as the design of our study. Thus, we opted for the use of a parametric test such as ANOVA. Furthermore, we wanted to predict a continuous outcome based on categorical predictor variables (i.e., Groups). Having more levels, a t-test would have given us little information that would not have allowed us to have a more complete view of the phenomenon as using ANOVA.

Reviewer #1, comment 4:
I suggest, after the Discussion section, enter the Conclusions section. This section is currently missing. Some of the data from Discussion should appear in Conclusions. I also propose to refer to future plans for this research.

Authors’ reply to Reviewer #1, comment 4:
We thank Reviewer 1 for their concern. We have now updated the manuscript by adding, after the discussion, a conclusion section in which we sum up our results and propose future lines of research: “In conclusion, our data show that the difficulty of an imagined action can influence valuation judgements. Moreover, our results support the hypothesis of tight relationship between motor and valuation processes. These links remain largely unexplored in current investigations, and they can be useful to better understanding some neurological or psychiatric conditions like schizophrenia, depression or frontal-apathy, which are characterized by an alteration of abilities to anticipate the effort necessary to obtain rewarding items in the environment [121–126]. Finally, future studies should also investigate the neural basis of this phenomenon; consequently, it would be very informative to submit a similar experimental paradigm to patients with focal brain lesions, for example lesions affecting the medial orbitofrontal cortex (mOFC) or the anterior cingulate cortex”.

Reviewer #1, comment 5:
The bibliography in the article is quite extensive, but often these are old items from 5 years ago. Some of them should be updated. For example, by referring to the article: refer to the article: Brain-computer technology-based training system in the field of motor imagery, IET Science Measurement and Technology from 2020

Authors’ reply to Reviewer #1, comment 5:
We thank Reviewer 1 for their questions and concerns. Accordingly, we have now updated our references with some new studies/reviews published in the past few years.

Reviewer 2 Report

In this manuscript, the authors tested the idea that imagined actions, just like real actions, affect how people assign value to a stimulus because they carry costs. The experimental task is appropriately designed and carefully conducted. The reviewer just has two concerns for the authors to address.

1, do the authors think 20 subjects each group is sufficient? did they perform a priori power analysis?

2, for data reporting, rather than simple bar plots showing the mean and sd, plotting individual variations (individual data points) together with the mean and sd is preferred.

Author Response

Manuscript ID: brainsci-1706944 – Sellitto et al., 

We thank the Editor for returning our manuscript following peer review. We appreciate the positive and constructive feedback of the two Reviewers. In the present revision, we have dealt with all the points raised, and we feel the manuscript has significantly improved as a result.

REVIEWER #2 – MINOR REVISIONS
In this manuscript, the authors tested the idea that imagined actions, just like real actions, affect how people assign value to a stimulus because they carry costs. The experimental task is appropriately designed and carefully conducted. The reviewer just has two concerns for the authors to address.

We thank Reviewer 2 for their appreciation of the quality of our work.

Reviewer #2, comment 1:
Do the authors think 20 subjects each group is sufficient? did they perform a priori power analysis?

Authors’ reply to Reviewer #2, comment 1:
We thank Reviewer 2 for giving us the opportunity to be clearer: We have now added in the manuscript the rationale behind the choice of the experimental sample size:
“A power analysis based on previously published studies (Vourvopoulos et al., 2017; Klement et al., 2018; Massar et al., 2020; Bowyer et al., 2021), indicated that a sample size of ~20 participants is necessary to achieve a statistical power of > 95% (2-tailed = 0.05).”

Reviewer #2, comment 2:
For data reporting, rather than simple bar plots showing the mean and sd, plotting individual variations (individual data points) together with the mean and sd is preferred.

Authors’ reply to Reviewer #2, comment 2:
We would like to thank the Reviewer for this suggestion, we now have modified the figures accordingly.
